# Transcriptome Analysis of Light-Regulated Monoterpenes Biosynthesis in Leaves of *Mentha canadensis* L.

**DOI:** 10.3390/plants10050930

**Published:** 2021-05-07

**Authors:** Xu Yu, Xiwu Qi, Shumin Li, Hailing Fang, Yang Bai, Li Li, Dongmei Liu, Zequn Chen, Weilin Li, Chengyuan Liang

**Affiliations:** 1Jiangsu Key Laboratory for the Research and Utilization of Plant Resources, Institute of Botany, Jiangsu Province and Chinese Academy of Sciences (Nanjing Botanical Garden Mem. Sun Yat-Sen), Nanjing 210014, China; yuxu@cnbg.net (X.Y.); xiwuqi@cnbg.net (X.Q.); shumingli@cnbg.net (S.L.); fanghailing@cnbg.net (H.F.); baiyang2020@cnbg.net (Y.B.); lili@cnbg.net (L.L.); dmeiliu@cnbg.net (D.L.); chenzq1219@cnbg.net (Z.C.); 2Institute of Botany, Jiangsu Province and Chinese Academy of Sciences, Nanjing University of Chinese Medicine, Nanjing 210023, China; 3College of Forest, Nanjing Forestry University, Nanjing 210037, China; wlli@njfu.edu.cn

**Keywords:** *Mentha canadensis* L., transcriptome sequencing, GC-MS, menthol biosynthesis

## Abstract

Light is a key environmental aspect that regulates secondary metabolic synthesis. The essential oil produced in mint (*Mentha canadensis* L.) leaves is used widely in the aromatics industry and in medicine. Under low-light treatment, significant reductions in peltate glandular trichome densities were observed. GC-MS analysis showed dramatically reduced essential oil and menthol contents. Light affected the peltate glandular trichomes’ development and essential oil yield production. However, the underlying mechanisms of this regulation were elusive. To identify the critical genes during light-regulated changes in oil content, following a 24 h darkness treatment and a 24 h recovery light treatment, leaves were collected for transcriptome analysis. A total of 95,579 unigenes were obtained, with an average length of 754 bp. About 56.58% of the unigenes were annotated using four public protein databases: 10,977 differentially expressed genes (DEGs) were found to be involved in the light signaling pathway and monoterpene synthesis pathway. Most of the TPs showed a similar expression pattern: downregulation after darkness treatment and upregulation after the return of light. In addition, the genes involved in the light signal transduction pathway were analyzed. A series of responsive transcription factors (TFs) were identified and could be used in metabolic engineering as an effective strategy for increasing essential oil yields.

## 1. Introduction

Light is one of the most important regulators of plant growth and development. It also has a significant impact on the differential accumulation of isoprenoid-derived metabolites [1]. Although significant research has been carried out on light-controlled seedling morphogenesis, seed germination, shade-avoidance, and flower induction [2,3,4,5], much less is known about how light regulates secondary metabolite biosynthesis, and about the regulatory mechanisms of monoterpenes biosynthesis. It has been reported that light intensity can alter terpenoids production via the activation of photosensitive enzymes involved in the methylerythritol phosphate (MEP) pathway [6]. Previous reports have shown that overexpressing Cryptochrome1 increased the amount of artemisinin in the medicinal plant *Artemisia annua* L. [7]. It has also been reported that the key transcription factor in the light signaling pathway, AaHY5, positively regulates the expression of genes involved in the artemisinin biosynthetic pathway [8].

*Mentha* L., belonging to the Labiatae family, is widely distributed and cultivated throughout the world. Because of their high economic value, *Mentha* species have become one of the most important essential oil crops in the world [9]. In peppermint, the biosynthesis of monoterpenes though the MEP pathway in plastids involves several characterized enzymes, including geranyl diphosphate synthase large subunit (GPPS-l), geranyl diphosphate synthase small subunit (GPPS-s), limonene synthase (LS), (−)-limonene-3-hydroxylase (L3OH), (−)-trans-Isopiperitenol dehydrogenase (iPD), (−)-isopiperitenone reductase (iPR), (+)-cis-isopulegone isomerase (iPI), (+)-pulegone reductase (PR), menthofuran synthase (MFS), and menthol dehydrogenase (MR). The relevant biosynthetic pathway is well understood, and all of the enzymes involved have been properly characterized [10]. Nevertheless, our understanding of the molecular mechanisms has revealed how the effects of the environment on monoterpenes biosynthesis are still unknown. Few transcription factors (TFs) have been reported to regulate monoterpenes biosynthesis. For instance, MsMYB negatively regulates monoterpenes production by binding to the *cis*-element of *MSGPPS.LSU* and suppressing its expression [11]. The overexpression of *MsYABBY5* was shown to lead to the increased production of terpenes [12], but their mechanisms of signaling conduction, and the mechanism of the response of their signaling to environmental factors, were elusive.

We investigated the effects of light on the densities of peltate glandular trichomes and on the chemical properties of the essential oil composition in mint leaves. High-throughput RNA-seq data were generated to screen expression profiles under the conditions of a 24 h darkness treatment and a recovery light period of 24 h. Differentially expressed genes (DEGs) of the light signal transduction pathway, as well as transcription factors and monoterpene biosynthetic genes, were identified, and their expression levels were compared. The results show that the expression of monoterpene biosynthesis genes is strongly downregulated by darkness, indicating their responsiveness to light. The results facilitate a further understanding of light signaling’s functional regulation of monoterpenoids biosynthesis. A series of identified responsive transcription factors (TFs) could be used in metabolic engineering to effectively increase essential oil yields.

## 2. Results

### 2.1. Low Light’s Effects on the Densities of Peltate Glandular Trichomes and Chemical Properties of Essential Oil Composition of Mint Leaves

We assessed the peltate glandular trichome densities on five separate zones, from the basal to the top regions, of the abaxial leaf surface of *Mentha canadensis* under different light conditions. The results showed that the peltate glandular trichome (PGTs) densities were reduced by half a percent in all five zones under the low light condition (Figure 1C). Basal zone 1 had a maximum of 27.75 ± 0.95 glands number per mm^2^, which decreased to 16.63 ± 2.67 glands per mm^2^ under the low light condition (Figure 1A,B). The top zone (zone 5) had a minimum PGT density of 11.5 ± 2.23 glands per mm^2^, which was significantly reduced to 4.87 ± 0.83 per mm^2^ under the low light condition (Figure 1A,B). An abundance of wrinkled PGTs were observed under low light conditions using scanning electron microscopy, indicating low essential oil accumulation in the subcuticular cavities of peltate glandular trichomes (Figure 1D).

The organic compounds volatilized from mint leaves were collected via headspace sorption and analyzed via GC-MS. Fifteen volatile compounds were identified, accounting for 99.87% and 97.13% of the total volatile exudates in mint leaves under control and low light conditions, respectively. The changes in total volatile compounds and monoterpene compositions were investigated. The relative contents of volatile compounds were dramatically reduced compared to the internal standard substance, which was bornanone. The level of the main oxygenated monoterpene, menthol (constituting 32.02% of total volatile compounds), decreased to 12.85% following low light treatment. The proportion of pulegone increased from 0.95 to 8.70%, and that of piperitone increased from 1.23 to 28.31% (Figure 2). The low light condition caused the accumulation of pulegone and piperitone, which are intermediate compounds in mint leaves, but had no reductive effect on the proportions of total monoterpene hydrocarbons or oxygenated monoterpenes (Table 1).

### 2.2. Illumina HiSeq mRNA Sequencing

To investigate the underlying role of light in menthol synthesis and identify genes with significantly different expression levels under dark conditions, we explored the transcription profiles of mint leaves in the dark for 24 h and light for 24 h, using RNA-seq.

The transcriptome sequencing was performed via an Illumina HiSeqTM 4000, which generated 72 million high-quality clean reads. Finally, a total of 95,579 all-unigenes with an N50 of 1260 bp were de novo assembled from the combined unigenes of three samples. The average length of the unigenes was 754bp (Appendix A). Raw sequencing reads were reported in the SRA database Available online: https://www.ncbi.nlm.nih.gov/Traces/study/?acc=PRJNA724910 (accessed on 28 April 2021)

Functional annotation of the 95,579 unigenes was carried out using five public protein databases. In total, 54,084 (56.58%) unigenes had a match in the Nr database, 32,399 (33.89%) in the swissprot protein database, 26,218 (27.43%) in the KOG database, and 46,665 (23.82%) in the KEGG database (Appendix A). For GO annotation, 64,497 unigenes (67.48%) were classified into 53 categories, in three parts: biological process (23,331), cellular component (21,172) and molecular function (19,994). In the “biological processes” category, the transcripts were enriched with the terms “metabolic” (18,732 unigenes), “cellular” (18,680 unigenes) and “single-organism” (15,645 unigenes). In the “cellular component” category, the assignments were mostly enriched with the terms “cell” (17,137), “cell part” (17,064) and “organelle” (14,479). In the “molecular function” category, “catalytic activity” (15,129), “binding” (12,910) and “transporter activity” were the most significantly enriched terms (Appendix A).

For the functional prediction, 26,219 unigenes were classified into 25 KOG categories. Among these, “General functional prediction only” (6420) was the largest group, followed by “Signal transduction mechanisms” (4527) and “Posttranslational modification, protein turnover, chaperones”, (2872) (Appendix A). Furthermore, 9690 unigenes were annotated in 138 KEGG pathways. “Metabolic pathways” (4150) was the most represented pathway (Appendix A).

### 2.3. Different Gene Expression Analysis

The expression profiles of *Mentha*
*canadensis* grown in darkness and light for 24 h each were analyzed separately and were selected based on two criteria: FDR < 0.05 and|log2FC| > 1. Compared to the samples under the control conditions, a total of 10,977 DEGs, including 6296 upregulated and 4681 downregulated genes, were differentially expressed after the 24 h darkness treatment. In total, we counted 8645 DEGs, including 3745 upregulated and 4900 downregulated genes compared to under darkness, after 24 h of light exposure (Figure 3). All DEGs were subjected to GO, KOG analysis, and KEGG pathway enrichment analysis. Based on the GO enrichment, “metabolic process” and “cellular process”, as well as “single-organism process”, were the three most highly represented terms in the “biological process” category. In the “cellular component” category, “cell”, “cell part” and “organelle” were the most enriched terms. In the “molecular function” category, most DEGs were enriched in the “catalytic activity”, “binding”, and “transporter activity” subcategories. The enrichment analysis showed that 1919 genes were annotated in 133 KEGG pathways. Among them, 108 genes were involved in the “metabolism of terpenoids and polyketides” pathway (Figure 4).

The expression profiles of all the DEGs adhered to two response patterns (Figure 5A,B). Seven significant expression clusters were observed in all the DEGs under darkness treatment followed by recovery light. The DEGs in cluster 5 were upregulated under darkness and downregulated after recovery (4736 genes), and those in cluster 2 were downregulated under darkness and upregulated under light exposure (3399 genes) (Figure 5C).

### 2.4. DEGs Involved in Terpenoids and Menthol Biosynthesis

From the transcriptome analysis, 14 DEGs involved in the monoterpenoids biosynthetic pathway were identified. These were terpene synthases (TPSs), (−)-isopiperitenone reductases (IPR), menthol dehydrogenases (MR), cytochrome P450 (P450), neomenthol dehydrogenases (NMRs) and alcohol dehydrogenase (ADH). The two TPSs were alpha-pinene/camphene synthase and linalool synthase. We identified 15 DEGs involved in sesquiterpenoids and triterpenoids. Excepted for the Unigene0029907 identified as (3S)-linalool synthase, the other 14 TPSs were dramatically downregulated in the dark but were then upregulated to the same expression level after regaining access to light for 24 h (Figure 6).

In mint, menthol was the most prevalent chemical compound among the monoterpenoids. In our study, orthologous genes encoding enzymes that catalyzed the biosynthesis of (−)-menthol were identified by multiple sequence alignment, using the reference genes of peppermint and spearmint. The nucleotides of these genes were cloned and sequenced in *M.*
*canadensis* (Appendix A). The expression profiles of six identified genes, including limonene synthase (LS), (−)-limonene-3-hydroxylase (L3OH), isopiperitenol dehydrogenase (iPR), isopiperitenone reductase (iPD), pulegone reductase (PR), and neomenthol dehydrogenase (MNR), showed the same patterns: they were significantly downregulated under 24 h of darkness and upregulated after returning to the light for 24 h (Figure 7). Only the MFS gene showed the opposite expression pattern, which was upregulation under darkness and downregulation after removing the covers. Although the GPPS-l, GPPS-s, and MR genes’ expression levels were not dramatically altered by light exposure, they were all downregulated as a result of darkness (Figure 7).

RT-PCR was used to verify the expression levels of menthol biosynthetic genes (Figure 8). The results indicate that the qRT-PCR expression patterns of 9 genes matched quite well with the RNA-Seq data.

### 2.5. Expression of Genes Involved in the Light Signal Transduction Pathway

The gene families of photoreceptors, such as phytochromes, cryptochromes, phototropins, and UVR8, were identified within the DEGs. The expression levels of PHYB (Unigene0004374) and PHOT2 (unigene0003571) were dramatically reduced in darkness and increased slightly when they re-entered light. The CRY gene (Unigene0004413) showed the opposite expression pattern: upregulated under darkness and downregulated after removing the covers. Other photoreceptors’ transcription expressions were changed similarly to those of the PHOTs, and these changes were not significant.

COP1, a ring-finger type ubiquitin E3 ligase, functions as a repressor of photomorphogenesis, and was significantly downregulated in darkness and upregulated when returning to light. HY5 and PIFs are the downstream components of photoreceptor-mediated light signaling. Both of the two HY5 potential homologs, and three potential homologs, were identified. HY5 was significantly downregulated under darkness and upregulated after removing the covers. PIFs are the key regulators in shade-avoidance responses. The transcript levels of PIF1 and PIF4 were dramatically upregulated following the 24 h darkness treatment, while PIF3 was not differentially expressed after the treatment (Figure 9).

### 2.6. Light-Mediated Regulation of Transcription Factors

Transcription factors play important roles in the response to light, and they regulate multiple genes. Using the PlantTFDB database, 1601 TFs belonging to 57 families were identified. After analyzing the differential expression profiles, totals of 380 and 298 DEGs divided into 44 TF families were differentially expressed. The bHLH, ERF, MYB, and MYB-related, C2H2, bZIP, WRKY, NAC, HD-ZIP, and GRAS families were the top 10 most prevalent TF families. Among them, 235 TFs were upregulated and 145 TFs were downregulated following the 24 h darkness treatment. After removing the covers for 24 h, 166 TFs were upregulated and 132 TFs were downregulated. There were 67 TFs showing similar expression patterns to the terpenoids synthesis pathway genes. The expression levels of these genes were downregulated by darkness and upregulated after returning to light. The MYB family was the largest group, followed by the bHLH family. The opposite expression pattern (upregulated in darkness and downregulated in light) was displayed in 137 TFs. Among them, the ERF family was the largest group (Appendix A).

Several transcription factors belonging to the bHLH, ERF, MYB, and WRKY families have been determined to control terpenoid biosynthesis. We identified the homologs of the TFs whose functions are known, and which regulate terpenoid biosynthesis, by BLAST searching our transcriptomic data, and identified two MYC1, one MYC2, one WIN1, one ERF1, one WRKY40 and one R2R3-MYB genes. Apart from ERF1, which was significantly upregulated by the darkness treatment, all the genes were downregulated. Interestingly, not all the downregulated genes were upregulated after 24 h light exposure. The transcription levels of the four TFs belonging to the bHLH family and one WIN1 gene were not changed after returning to light for 24 h. The transcription levels of one WIN1, one WRKY40 and one R2R3-MYB were differentially expressed: the WRKY40 was upregulated back to its previous level before the darkness treatment, while the WIN1 and R2R3 MYB remained lower than their initial levels (Figure 9).

### 2.7. Validation of Expression Patterns of Genes

To verify the reliability of the RNA-Seq data, the expression profiles of eight genes were analyzed using quantitative real-time PCR (qRT-PCR). These genes are listed in Appendix A. The expression patterns of the selected genes, in terms of the RPKM results, were consistent between the qRT-PCR and RNA-seq (Figure 10). The following validation results for qRT-PCR indicate that the RNA-Seq data are quite reliable.

## 3. Discussion

Glandular trichomes, referred to as true cell factories, are able to secrete and store large quantities of specialized metabolites. The monoterpenes of mint essential oil are produced by, and stored in, the peltate glandular trichomes [13]. The number of glandular trichomes per leaf, and their production of chemical compounds, depend on environmental and developmental parameters [14]. The distribution of peppermint gland initiation reflects the basipetal pattern of leaf maturation: relatively immature regions at the leaf’s base continue to produce oil glands long after gland production has stopped at the leaf’s apex [13]. To accurately determine the densities of mint glandular trichomes under low light conditions, five different areas at the same stage of development, from the basal zone to the top zone of an abaxial leaf, were analyzed. Our results show remarkably reduced densities of peltate glandular trichomes in all five selected leaf zones under low light conditions, which correlates with reduced chemical compound production. The present results suggest that reduced light intensity effects not only peltate glandular trichome density, but also the production and accumulation of essential oil, in *Mentha*
*canadensis*. This is consistent with the results for *Mentra piperita* and *Mentha arvensis* [15,16]. In other plant species, cultivated tomato for instance, increases in photosynthetically active radiation induce the growth of type VI leaf glandular trichomes [17]. In the case of *Varronia curassavica* Jacq., higher irradiance results in an increased essential oil yield and a higher frequency of glandular trichomes [18]. The conclusion is that light density regulates not only the density of mint’s glandular trichomes but also the synthesis of essential oil. However, the molecular mechanisms by which light regulates monoterpenes biosynthesis, as well as the reprogramming of the pathway, are still not fully understood. To study the process in more detail, the mRNA expression patterns under changing light conditions were assessed via transcriptome analysis.

Transcriptome sequencing is a powerful tool for screening gene expression patterns and identifying candidate genes. In the current study, high-throughput RNA sequences were generated to analyze the transcriptomes of *M. canadensis* undergoing a 24 h darkness treatment followed by exposure to light for 24 h. Approximately 72 million high-quality clean reads were generated, and a total of 95,579 unigenes, with an N50 of 1260 bp and average length of 754 bp, were obtained. In a previous study, nine of the orthologous genes involved in the menthol synthesis pathway were identified. Darkness for 24 h downregulated the expression levels of *McGpps-L*, *McGpps-s*, *McLs*, *McL3OH*, *MciPD*, *MciPR,* and *McPR* genes. The RNA-Seq study showed that the expression levels of these genes were upregulated after light exposure, and RT-PCR confirmed these results. Among the transcriptional profiles, 14 of the DEGs involved in the monoterpene pathway and 15 of the DEGs involved in the sesquiterpenoid and triterpenoid biosynthesis pathways showed the same transcription patterns, except for unigene0051735 and unigene001095. The significant downregulation of genes involved in the MEP pathway may explain the reduced essential oil metabolite production. Furthermore, changes in the monoterpene profiles show that low light intensity leads to an accumulation of the branchpoint intermediate pulegone and the side product piperitone. In fact, under conditions of salinity and osmotic stresses, pulegone accumulated in *Mentha* species [19,20,21,22]. This intermediate compound is considered an undesirable component under harsh growth conditions. The downregulation of the gene encoding pulegone reductase (PR), and the consistent expression of the gene encoding (−)-menthol dehydrogenase (MR), may explain the reduced proportions of menthone and menthol. Mahmoud and Croteau [23] reported that high levels of menthofuran reduced pr transcript levels in immature leaves, consequently increasing pulegone content in peppermint. These results suggest that the flux of (+)-pulegone is controlled at the transcriptional level of pr, and they are consistent with the results of our present research. However, our data are presently insufficient to explain the accumulation of piperitone under low light conditions.

The light-dependent transcriptional regulation of MEP pathway genes is conserved in all plant species we investigated. Nevertheless, not much is known about the signal transduction pathways or the transcription factors that convert the environmental information into a transcriptional response [1]. In our transcriptome analysis results, co-expression modules with terpenoid biosynthesis genes from the MEP pathway suggest that the transcription of those genes may be coregulated by the same transcription factor as response to light. In *A. annua*, HY5, a bZIP transcription factor and a core TF in the light signaling pathway, interacts with the promoter of the monoterpene synthesis gene *QH6* [24]. It has also been reported that AaHY5 acts via trichome-specific and light-induced AaGSW1 to control artemisinin formation [8]. Evidence from *Arabidopsis* has demonstrated that HY5 and PIFs are direct regulators of the light-modulated expression of the *DXS* and *DXR* genes encoding the flux-controlling enzymes of the MEP pathway [25]. In the present transcriptome data, homologous genes of *HY5* (Unigene0034124), *PIF1* (Unigene0086430) and *PIF4* (Unigene0065109) were identified in *M. Canadensis*. The resultant RPKM values indicate that the expression levels of these TF genes were significantly changed under dark conditions. The expression pattern of *HY5* was the same as that of the menthol synthesis genes, whereas *PIFs* showed the opposite expression pattern (upregulated under darkness and downregulated after illumination). These results indicate that a similar regulation mechanism to that targeting the monoterpene synthesis gene (operating via the key-master integrators of light signal transduction HY5 and PIFs) may also operate in *M. canadensis*. More evidence should be sought in future experiments.

We have established strategies to increase the levels of essential oil in peppermint via the overexpression of unique structural genes in the menthol synthesis pathway, or the overexpression of additional genes that encode enzymes involved in the precursor supply pathway for monoterpene biosynthesis [26]. Transgenic lines were generated, but few of them improved essential oil yields. The overexpression of transcription factors that target and positively control multiple monoterpene biosynthesis, or the suppression of the expression of negatively controlled TFs, could also each represent metabolic engineering strategies for improving essential oil yields. Previously, researchers have elucidated the regulatory roles of several genes encoding the WRKY [27,28,29], MYB [30], bHLH [31], AP2/ERF [32,33], and bZIP [8] classes of TFs in specialized terpenoid biosynthesis. Using RNA-Seq in *Solanum lycopersicum*, SlMYC1 was identified and shown to transiently transactivate the terpene synthase promoters in *Nicotiana benthamiana* leaves [34]. Heterologous AtWRKY18, AtWRKY40, and AtMYC2 were overexpressed in *S. sclarea* hairy roots and proven to regulate the expression of several genes encoding enzymes of the MEP-dependent pathway [35]. The overexpression of SmERF128 increased the expression levels of copalyl diphosphate synthase 1 (SmCPS1), kaurene synthase-like 1 (SmKSL1) and cytochrome P450 monooxygenase [32]. The functional study of TFs showed that the regulated monoterpene genes in mint were inadequate. The TFs MsYABBY5 and MsMYB were reported to negatively regulate the production of monoterpenes in spearmint [11,12]. In our transcriptome data, Unigene0018692 was homologous to MsMYB, and was found to be dramatically differently expressed under dark conditions. Interestingly, given that it was regulated by light, the expression pattern of this gene was similar to that of the negatively regulated genes, which were downregulated by dark and upregulated by light (mean RPKM value of 24.78 in control vs. 0.76 under darkness and 5.69 after illumination). These results suggest that other TFs must be more essential in regulating monoterpene gene expression through light signal transduction pathways. Our present work provides a list of candidate genes for further metabolic engineering research and will be helpful in exploring the molecular mechanism of the transcriptional responses of MEP pathway genes to light signals. However, the present RNA-seq results could not determine whether splice variants could vary between the absence and presence light. Full-length transcriptome analyses and more verification research should be carried out in the future.

In conclusion, the results of this study prove the effects of light on the density of peltate glandular trichomes and essential oil production in cornmint. The low light condition reduced the PGT densities in all five different zones, from the bottom to the top of the mint leaves, by half a percent. We observed dramatically decreased total essential oil and menthol levels, and increased proportions of pulegone (from 0.95 to 8.70%) and piperitone (from 1.23 to 28.31%). Downregulated expression levels of menthol synthesis pathway genes McGpps-L, McGpps-s, McLs, McL3OH, MciPD, MciPR, and McPR and the monoterpenes biosynthesis genes, as well as the sesquiterpenoids biosynthesis pathway genes, were observed following darkness treatment, and may explain the reduced levels of essential oil produced via the regulatory mechanisms at the transcription level. The identified DEGs in the light signal transduction pathway, and the light-mediated regulation of transcription factors, could be used as effective strategies to achieve increased essential oil yields in future metabolic engineering studies.

## 4. Materials and Methods

### 4.1. Plant Materials and Treatment

For phenotype difference experiment, mint (*Mentha* *canadensis*) plants were grown on the field in Institute of Botany, Jiangsu Province and Chinese Academy of Sciences, Nanjing, Jiangsu Province, China (28°05′ N, 113°37′ E). Low-light treatment group was performed by covering plots with sunshade nets at a height of 2.5m above the ground, which caused a reduction of 50% light. The control group grown at normal Sunshine. Two groups both grown in the same soil with the same 12 h photoperiod. In total, three replications were made for the control and the low light groups, respectively, each replication containing 10 plants. After 30 d low light treatment, leaves were harvest to statistic of peltate glandular trichomes densities and analysis essential oil composition.

For transcriptome analysis, mint (*Mentha Canadensis*) were propagated from rhizomes and were planted in plastic pots containing a mixture of soil and vermiculite (3:1, *v*/*v*). Plants were cultured in growth chamber under control condition at 1000 μmol·m^−2^·s^−1^, with the 14 h photoperiod and temperature maintained at 25 °C/22 °C. For darkness treatment, black paper bags was used to cover the whole plants for 24h, then move the bags and exposure the plants to light 24 h. Leaves at the same developing stage were collection at 24h darkness treatment time point and at 24 h illumination time point. Control materials were collected at the same time point as darkness treatment. Three plants were selected as biological replicates. Samples were frozen in liquid nitrogen and stored at −80 °C before RNA extraction.

### 4.2. Statistics of Peltate Glandular Trichomes Densities and Measurement of Essential Oil Composition in Mint Leaves under Darkness Treatment

Leaves at the same development stage were divided into five sampling zones from bottom to top. The number of peltate glandular trichomes in each 1 mm^2^ were counted after taking a photograph using fluorescence stereomicroscope MVX 10. The analysis was performed in fifth repetition.

Sampling of volatiles from 0.5 g dry weight leaves were collected by headspace solid-phase microextraction (HS-SPME), followed by gas chromatography/mass spectrometry (GC/MS) determination. The polydimethylsiloxane (PDMS) 65 µm fiber (65 µm Carboxen™-PDMS StableFlex, Supelco) was then exposed to the headspace of the sample for 3 min at 60 °C to adsorption. Then volatiles absorbed were desorbed in the injection port of GC at 250 °C for 3 min. Each sample was spiked with 300 ng bornanone internal standard. The analysis was performed in triplicate.

GC analyses were carried out using a Trace 1300 gas chromatograph equipped with a flame ionization detector (FID) and an electronic pressure control (EPC) injector. GC setting were as follow: inlet 250 °C, the carrier gas was N2 (U) with a split flow 50.0 mL/min and the split ratio was 33:1. Analyses were performed using the following temperature program: oven temperature isotherm at 60 °C for 1 min, from 60 °C to 220 °C at the rate of 3 °C min^−1^.

GC-MS analysis was performed on a gas chromatograph Trace 1300HP interfaced with a HP 5972 mass spectrometer with electron impact ionization (70 eV). The column temperature was programmed as follows: initial temperature was 50 °C, increased to 240 °C at 5 °C min-1, and isotherm at 205 °C during 10 min. Helium was used as the carrier gas with a flow rate of 1.0 mL/min^−1^. The split ratio was 33:1. Scan time and mass range were 0.2 s and 33–450 *m*/*z*, respectively.

Identification of volatile compounds was based on the comparison of their mass spectral fragmentation patterns with the National Institute of Standards and Technology (NIST) mass-spectral library data and other published mass spectra (Adams 2001). The percentage content of compounds was calculated by the area normalization method.

### 4.3. Total RNA Extraction, Library Construction and Transcriptome Sequencing

Total RNA from nine samples including three repetition samples of control, 24 h darkness treated and 24 h exposure to light leave materials were isolated using RNA iso Plus reagent Kit (Takara Bio, Dalian, China). First strand cDNA was synthesized using M-MLV Reverse Transcriptase (Promega, Madison, WI, USA). The cDNA library was constructed and transcriptome sequencing was performed by ORI-GENE Company (Beijing, China). The transcriptome libraries were sequenced using an Illumina HiSeq™ 4000 (Illumina, Inc., San Diego, CA, USA) at the Gene De novo Institute (Guangzhou, China).

### 4.4. De Novo Assembly and Functional Annotation

After removing the adaptor and low-quality sequences, a de novo assembly of the remaining high-quality clean reads was performed using Trinity, as previously described for use without a reference genome [36]. After assembly, unigenes were obtained and functional annotation was carried out using BLASTx program (Available online: http://www.ncbi.nlm.nih.gov/BLAST/) (accessed on 30 April 2021) with an E-value threshold of 1e−5. Public databases including the NCBI non-redundant protein database (Nr) (Available online: http://www.ncbi.nlm.nih.gov) (accessed on 30 April 2021), the SwissProt database (Available online: http://www.expasy.ch/sprot) (accessed on 30 April 2021), the Kyoto Encyclopedia of Genes and Genomes database (KEGG) (Available online: http://www.genome.jp/kegg) (accessed on 30 April 2021), and the COG/KOG database (Available online: http://www.ncbi.nlm.nih.gov/COG) (accessed on 30 April 2021), were used to annotate the M. canadensis unigenes. TFs were identified by aligning unigenes to the PlantTFDB database [37].

### 4.5. Identification and Analysis of Differentially Expressed Genes

The expression levels of the unigenes were normalized to RPKM [38]. The DEGs were defined as having an absolute fold-change value ≥2 and a false discovery rate (FDR) < 0.05 and |log2FC| > 1. Heatmaps of DEGs were generated using HemI [39].

### 4.6. Real-Time Quantitative PCR Analysis

qRT-PCR was performed to verify the expression of selected genes. First-strand cDNA was synthesized with oligo (dT)18 and M-MLV reverse transcriptase (Promega). qRT-PCR analysis was carried out using the SYBR Universal qPCR Kit (Vazyme) on a qTOWER2.2 Real Time PCR Systems (Analytik, Jena, Germany), according to the manufacturer’s instructions, and subjected to qPCR as described in Qi, et al. [40]. Quantification was performed using the 2^−ΔΔCT^ method, and data were normalized to those of the actin gene transcript. Sequences of primers used are listed in additional file Appendix A. RT-PCR analysis was conducted with three technical replicates, and the data represent the means ± standard errors (n = 3).

## Figures and Tables

**Figure 1 plants-10-00930-f001:**
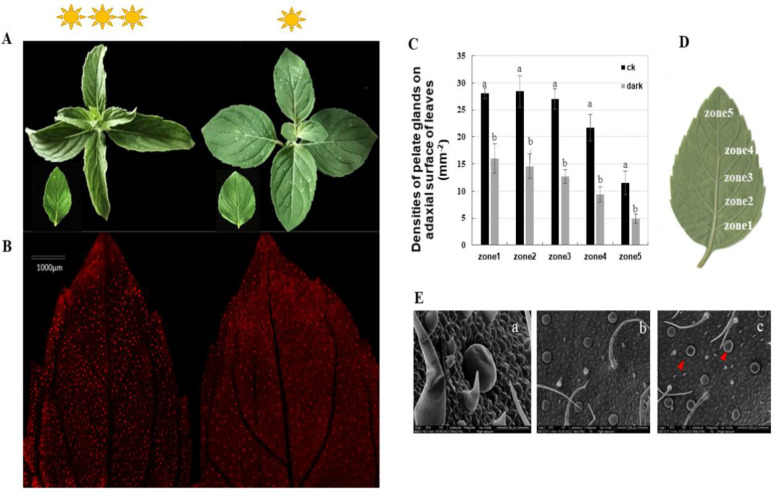
Phenotypes of *Mentha*
*canadensis* after 4 weeks of growth under low light conditions. (**A**) The morphology of leaves at the same developmental stage under normal light conditions (left) and low light conditions (right). (**B**) The abaxial surface of leaves under fluorescence light. Red spots show the spontaneous fluorescence of peltate glandular trichomes. (**C**,**D**) Peltate glandular trichome densities of the abaxial surface—five leaf zones from basal to top regions of young leaves. Bars indicate the densities of peltate glandular trichomes. Error bars represent SD. (**E**) SEM micrographs showing PGTs on the abaxial leaf surface. (**a**) Morphology of peltate glandular trichomes and non-glandular trichomes; (**b**) abaxial leaf surface under normal light condition; (**c**) abaxial leaf surface under low light growth conditions; the red triangles indicate wrinkled PGTs. The analysis of trichome densities was repeated five times.

**Figure 2 plants-10-00930-f002:**
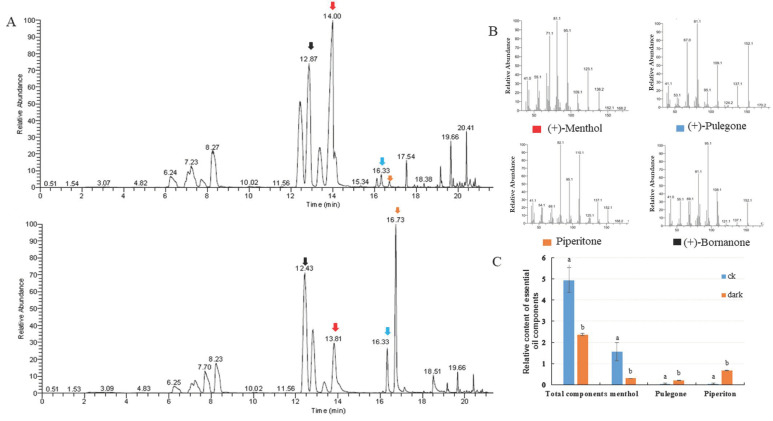
Introduction of changes in volatile exudates of leaves under the control and the low light conditions. (**A**) Gas chromatography–mass spectrometry (GC-MS) analysis of plants under the control (up) and low light (bottom) conditions. (**B**) MS/MS spectra of significantly changed compounds between the two light conditions, i.e., menthol, pulegone, piperitone, and the internal standard substance bornanone. (**C**) Relative contents of volatile compounds: menthol, pulegone, and piperitone.

**Figure 3 plants-10-00930-f003:**
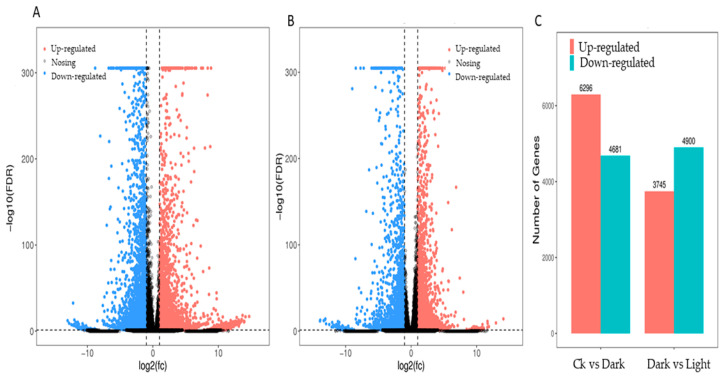
DEG statistics under controlled light and darkness conditions in *M. canadensis* transcriptomes. (**A**) Volcano plots of total DEGs in the leaves’ expression profiles after 24 h of darkness treatment. (**B**) Volcano plots of total DEGs in the leaves after 24 h of light exposure. (**C**) Total DEGs in the leaves’ expression profiles after darkness treatment and light exposure.

**Figure 4 plants-10-00930-f004:**
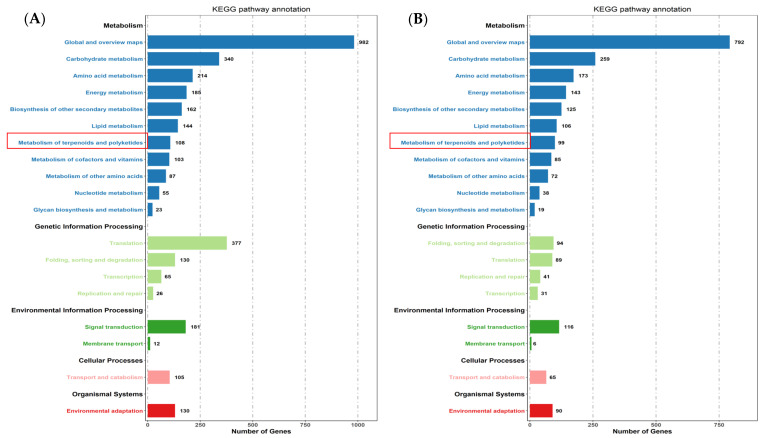
KEGG pathway enrichment of DEGs after controlled light vs. 24 h darkness treatment (**A**) and controlled darkness vs. 24 h light exposure (**B**) in *M. canadensis* leaves’ transcriptomes.

**Figure 5 plants-10-00930-f005:**
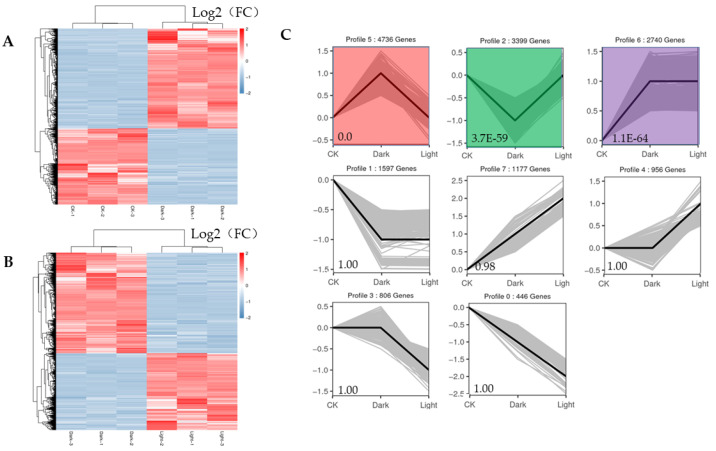
Heatmaps of all DEGs and eight significant expression clusters of all DEGs. (**A**) Heatmap showing the expression profiles of all DEGs between the control and darkness treatments. (**B**) Heatmap showing the expression profiles of all DEGs between the darkness and recovery light treatments. (**C**) The eight expression patterns of DEGs; the cluster names and unigene numbers are shown at the top.

**Figure 6 plants-10-00930-f006:**
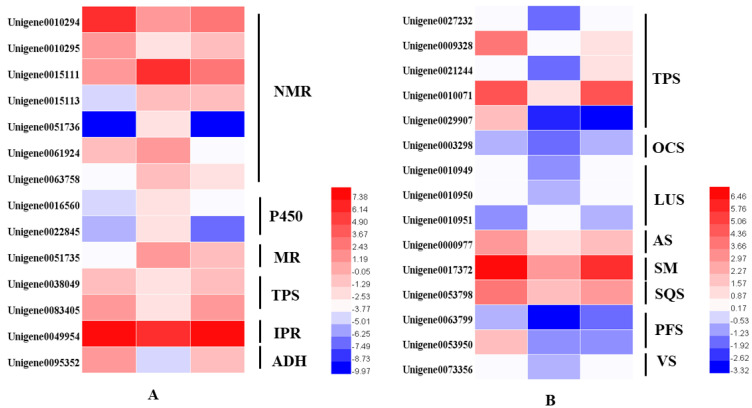
Heatmaps of the DEGs in the monoterpenoid biosynthesis pathway (**A**) and in the sesquiterpenoid and triterpenoid biosynthesis pathways (**B**). NMR, (+)-neomenthol dehydrogenase; P450, cytochrome; MR, (−)-menthol dehydrogenase; TPS, terpene synthase; IPR, (−)-isopiperitenone reductase; ADH, alcohol dehydrogenase.

**Figure 7 plants-10-00930-f007:**
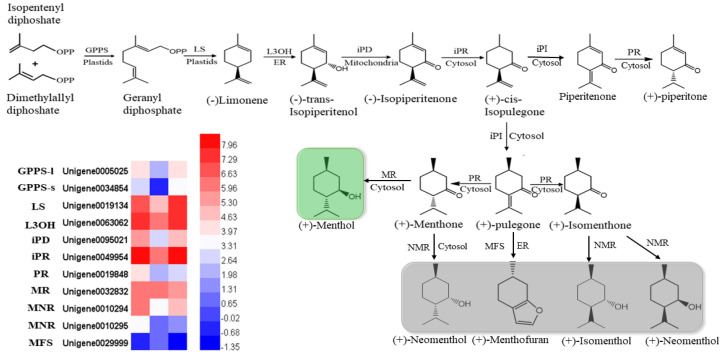
Heatmaps of the DEGs in the monoterpenoid biosynthesis pathway. GPPS-l, geranyl diphosphate synthase large subunit; GPPS-s, geranyl diphosphate synthase small subunit; LS, (−)-limonene synthase; L3OH, (−)-limonene-3-hydroxylase; iPD, (−)-trans-Isopiperitenol dehydrogenase; iPR, (−)-isopiperitenone reductase; iPI, (+)-cis-Isopulegone isomerase; PR, (+)-pulegone reductase; MFS, menthofuran synthase; MR, (−)-menthol dehydrogenase. The green block show the final important compound, menthol; gray blocks show the branches of the final compounds.

**Figure 8 plants-10-00930-f008:**
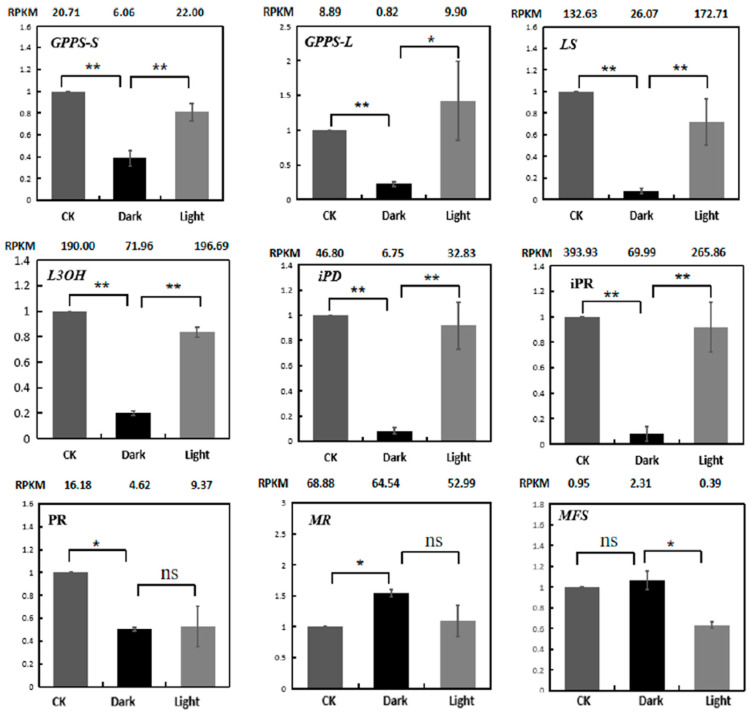
Expression validations of nine menthol synthesis genes in the control, darkness and recovery light samples using qRT-PCR. The RPKM values obtained from RNA-Seq data are shown at the top of each figure. The level of significance, obtained with Student’s *t*-test, is denoted as follows: * *p* < 0.05; ** *p* < 0.01. ns, not significant. Each value is the mean of at least three replicates.

**Figure 9 plants-10-00930-f009:**
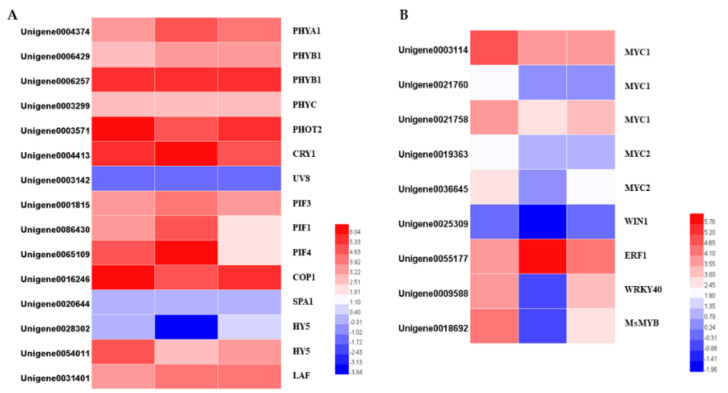
Heatmaps of the DEGs in the light signal transduction pathway (**A**) and heatmaps of differentially expressed transcription factors (**B**).

**Figure 10 plants-10-00930-f010:**
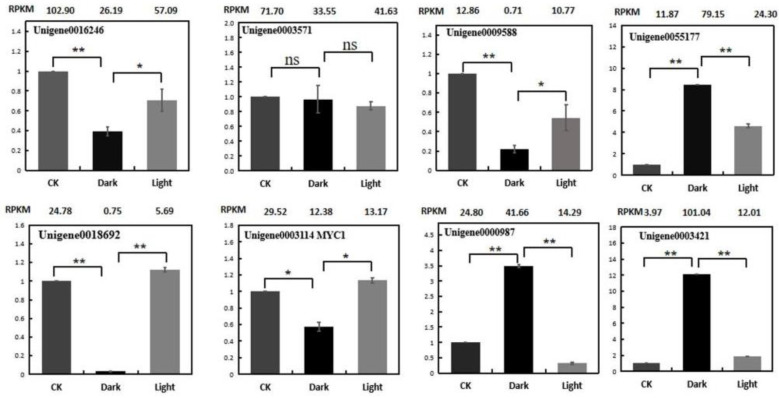
Expression validations of eight selected genes in the control, darkness and recovery light samples using qRT-PCR. The RPKM values obtained from the RNA-Seq data are depicted at the top of each figure. The level of significance, determined via Student’s *t*-test, is depicted as follows: * *p* < 0.05; ** *p* < 0.01. ns, not significant. Each value is the mean of at least three replicates.

**Table 1 plants-10-00930-t001:** The essential oil compositions of mint leaves (*Mentha canadensis* L.) under the control and low light conditions.

NO.	Compounds	Percentage of Total Monoterpenes +/−SD		Fold Change Low Light/Control Light
		Control Light	Low Light	
1	(+)α-Pinene	4.07 ± 0.61	3.52 ± 0.59	0.86
2	Sabinene	0.24 ± 0.03	0.12 ± 0.03	0.50
3	β-Pinene	1.05 ± 0.15	0.93 ± (0.08)	0.88
4	β-Myrcene	2.35 ± 0.14	1.90 ± (0.35)	0.81
5	Limonene	10.49 ± 1.29	11.07 ± (1.84)	1.05
6	1,8-Cineole	0.17 ± 0.02	0.17 ± 0.01	1.00
7	Menthone	25.18 ± 0.06	17.97 ± 0.28	0.71 *
8	Isomenthone	7.52 ± 0.49	3.12 ± 0.05	0.41 *
9	Menthol	32.02 ± 4.69	12.85 ± 0.41	0.40 *
10	Isopulegone	1.91 ± 0.15	0.15 ± 0.02	0.08 *
12	Pulegone	0.95 ± 0.22	8.70 ± 1.15	9.15 **
13	Piperitone	1.23 ± 0.63	28.31 ± 1.14	23.02 **
14	Caryophyllene	0.50 ± 0.06	0.16 ± 0.03	0.32 *
15	Germacrene D	2.9 ± 0.38	0.15 ± 0.02	0.16 *
16	γ-Cadinene	0.26 ± 0.02	0.26 ± 0.02	1.00
others		14.38 ± 2.05	18.52 ± 0.27	1.29 *
Monoterpene hydrocarbons		17.71 ± 2.31	16.66 ± 2.26	0.94
Oxygenated monoterpenes		67.67 ± 5.22	64.87 ± 2.24	0.95
Sesquiterpenes		10.07 ± 1.38	5.20 ± 0.73	0.51 *
Total essential compound percentage		99.87 ± 0.08	97.13 ± 0.27	0.97

The percentages of total monoterpenes are given as mean values ± SD. The fold change was calculated by dividing the percentages of compounds in darkness-treated leaves by the percentages of compounds in control leaves. The level of significance was obtained with Student’s *t*-test, and is marked as follows: * *p* < 0.05; ** *p* < 0.01.

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
