# Peer review of "Transcriptome Analysis of Light-Regulated Monoterpenes Biosynthesis in Leaves of Mentha canadensis L."

_plants, 2021, doi:10.3390/plants10050930_

Round 1
Reviewer 1 Report
The manuscript " Transcriptome analysis of light-regulated monoterpenes bio-synthesis in leaves of Mentha canadensis L." by Yu et al. (plants-1116966) is a very interesting work concerning the analysis of peltate glandular trichomes in the leaves of mint and the gene expression related to the expression of essential oils in this economically relevant plant. The work is well designed and a significant amount of data is provided. Nonetheless, I believe that some changes should be made before recommending the publication of the paper.
Despite the significant improvement of the manuscript in relation to its previous version, English grammar should still be amended. There is a great number of examples of lack of grammatical concordance between subject and verb, as well as many missing particles in the text. Just a couple of examples:
- Line 73-74: "The monoterpenes of mint essential oil is produced..."
- Line 177: "The enzymes for each catalyzing steps was isolated..."
I won´t go into more detail, the text should be thoroughly checked and corrected.
Issues with scientific language:
- Scientific names of the species should always be in italics, and the second part of the name (species) must not start with a capital letter.
- The first time a species name is used it should be written in full.
- Acronyms should be explained the first time they are used (line 39 : MEP).
Other issues:
- Line 94-95: Low light might induce the accumulation, but it does not accumulate.
- Line 112: What is "aerial parts of mint"?
- Why do the authors use Standard error in some graphs but Standard deviation in others?
- Line 124: 540849 unigenes. Please correct.
-In some cases a comma is used to specify thousands, but in others not. Please use this consistently.
-Line 139-140: That is not what can be seen in Figure S3. Besides, caption in the Figure says Figure 4.
- Line 178: cDNAs from spearmint: this species is not mentioned in M&M.
- Lines 184-185: 15 TPSs, do the authors mean DEGs? Besides, if one is not then is 14 not 15.
- Figure 7: Legend does not explain the content of the Figure.
- Figure 8: legend should explain the content as in Figure 10. On the other hand, a representation of the RPKM expression using a line and the right axis of the Figures should have been more accurate for comparison.
- Supplemental Figure 4: the same color code should be used for each TF family in both representations, for clarity purposes.
-Line 317: Mahmoud and Croteau.
-Line 318: "decrease of pr message"?
- Some expressions that do not make sense:
- Line 232: "...PIF1 and PIF4 were in resonse to darkness..."
- Line 361.
- Lines 398-399.
In conclusion, I encourage the authors to improve the English in this manuscript and make the changes suggested in order o publish the significant results they obtained.
Author Response
Dear reviewer:
We are grateful to reviewers’ critical comments and thoughtful suggestion. Based on these comments and suggestion, we have made careful modifications on the original manuscript. We rearranged sentences in article and improved the figures, In addition, we check and revised the reference. We hope the new manuscript will meet your magazine’s standard. Here we did not list all the changes but marked in red in revised paper. We appreciate for Editors’ warm work earnestly, and hope that the correction will meet with approval.
We rearranged sentences in article and improved the figures,We checked all the scientific names and corrected all the names.
- Acronyms should be explained the first time they are used (line 39 : MEP).
We revised the MEP to methylerythritol phosphate
Other issues:
- Line 94-95: Low light might induce the accumulation, but it does not accumulate.
We revised this sentences.
- Line 112: What is "aerial parts of mint"?
We changed the aerial parts to leaves
- Why do the authors use Standard error in some graphs but Standard deviation in others?
It should be standard deviation in all the graphs.
- Line 124: 540849 unigenes. Please correct.
-In some cases a comma is used to specify thousands, but in others not. Please use this consistently.
We changed all the numbers
-Line 139-140: That is not what can be seen in Figure S3. Besides, caption in the Figure says Figure 4.
It should be figure 3 in Figure S3
- Line 178: cDNAs from spearmint: this species is not mentioned in M&M.
We delete the discussion part in the results.
- Lines 184-185: 15 TPSs, do the authors mean DEGs? Besides, if one is not then is 14 not 15.
Yes, it is DEGs, we changed the number, it should be 14.
- Figure 7: Legend does not explain the content of the Figure.
We revised the Figure 7 legend.
- Figure 8: legend should explain the content as in Figure 10. On the other hand, a representation of the RPKM expression using a line and the right axis of the Figures should have been more accurate for comparison.
We revised the Figure 8 Legend.
- Supplemental Figure 4: the same color code should be used for each TF family in both representations, for clarity purposes.
We revised the Supplemental Figure 4 and change the same TF family as the same color code.
-Line 317: Mahmoud and Croteau.
Revised
-Line 318: "decrease of pr message"?
Revised the sentence.
Thank you and best regards.
Yours sincerely,
Dr Chengyuan Liang
Reviewer 2 Report
This study investigates the transcriptome of light-regulated monoterpene biosynthesis in Mentha canadensis leaves. The work is interesting and well-written; however, I have some observations that are listed in the following lines.
- Throughout the manuscript, avoid starting a sentence with a number that is not written out.
- Line 39: methylerythritol phosphate (MEP)-pathway
- Lines 73, 75, 219, and 231: Avoid using references in the results section. The results should describe the observations and focus on the findings of the current study only. Rewrite this section.
- Figure 4 is not readable.
- Add a section for the conclusions.
- Line 421: use the same reference style for all the references.
Author Response
Dear editor and reviewer:
We are grateful to reviewers’ critical comments and thoughtful suggestion. Based on these comments and suggestion, we have made careful modifications on the original manuscript. We rearranged sentences in article and improved the figures, In addition, we check and revised the reference. We hope the new manuscript will meet your magazine’s standard. Here we did not list all the changes but marked in red in revised paper. We appreciate for Editors’ warm work earnestly, and hope that the correction will meet with approval.
- Throughout the manuscript, avoid starting a sentence with a number that is not written out.
We checked and revised.
- Line 39: methylerythritol phosphate (MEP)-pathway
We rewrited it .
- Lines 73, 75, 219, and 231: Avoid using references in the results section. The results should describe the observations and focus on the findings of the current study only. Rewrite this section.
We rewrite this section and deleted all the references in this part.
- Figure 4 is not readable.
We changed a visible figure
- Add a section for the conclusions.
We added the conlusions section
- Line 421: use the same reference style for all the references.
We checked the references style.
Reviewer 3 Report
The authors test the hypothesis how light affect the production of certain compounds with an emphasis on oils. As the observe a different in the production by applying low light production or darkness to the plants and using GC-MS could monitor a quantitive difference between the two states. Next, the apply RNA-seq to monitor differential gene expression between the two states. They identify different genes that are regulated and expressed differently and test their hypothesis using qRT-PCR.
I have some minor and major concerns here.
Minor :
language should in general be improved for this paper.
also the formating of institutions needs to be formatted correctly. "he basal zone1 had the maximum of 27.75 ± 0.95 glands 81number per mm2 which decreased to 16 ± 2.67" -> 16.00 ? please be consistent with significant numbers - it means different things. Line 121-132 if it is important include figure into main text - the text is very hard to understand. Figure 1 please add N=? Figure 4 - poor quality. very hard to read. Figure 8 - in statistics there are clear way to annotate p-values please use those instead of inventing new annotations. Table 1 - hard to understand how things were compared what is a and b. what was the null hypothesis. The reviewer is a a bit uncertain here - in figure 1 you use SEM and here it is SD if correct state why. for the statistics please use convention in the figures - the a,b etc is very hard to grasp for readers.Major concern:
My major concern is around the RNAseq experiment. Firstly, for the comparison in the figures it would be much better and more correct to display the TPM value instead of the RPKM for the direct comparison between experiments.
No power analysis has been performed for the study and it is hard to know the detect limiting of the differential expressed genes from this study.
Furthermore, there is a huge variability in RNA seq but no technical or biological replicates has been performed to reduce noise in assay.
Also looking at the RPKM versus qPCR numbers it is in many cases hard to any correlation between the numbers if though that is stated. The figures are hard to understand and should be rethought. Again it would be interesting to see how the TPM correlates with this.
It is only not clear to this review if splice variants could play a role between plus/minus light - this study cannot address this but a comment to the limitations in the current study should be address in the discussion.
Overall the idea is very interesting but the RNA seq study is not convincing in the current form. More replica needs to be performed to draw the conclusion wanted from the researchers.
Author Response
Dear editor and reviewer:
We are grateful to reviewers’ critical comments and thoughtful suggestion. Based on these comments and suggestion, we have made careful modifications on the original manuscript. We rearranged sentences in article and improved the figures, In addition, we check and revised the reference. We hope the new manuscript will meet your magazine’s standard. Here we did not list all the changes but marked in red in revised paper. We appreciate for Editors’ warm work earnestly, and hope that the correction will meet with approval.
We revised the format. We revised the table 1 and changed the a,b to * in the figures. We changed a high quality figure instead of figure 4.
We caculated the TPM and listed the TPM versus RPKM in the table below, we found that the RPKM value is quite similar with TPM. The verfified genes involved in the menthol synthesis pathway has the same expression level change patterns after using TPM value. We found several references[1-3] that also use RPKM for the comparison between experiments. We generated three technical replicates to our RNA-seq datas and each RPKM value was the mean of three replicates.
Once again, thank you very much for your comments and suggestions.
Table 1 TPM versus RPKM values about the genes involved in the menthol synthesis pathway
|
|
CK |
|
Dark |
|
Light |
|
|
|
Mean TPM |
Mean RPKM |
Mean TPM |
Mean RPKM |
Mean TPM |
Mean RPKM |
|
GPPS-L |
9.87 |
8.89 |
0.86 |
0.82 |
10.82 |
9.9 |
|
GPPS-S |
22.04 |
20.71 |
5.95 |
6.06 |
23.09 |
22 |
|
LS |
129.53 |
132.63 |
23.95 |
26.07 |
166.26 |
172.71 |
|
L3OH |
193.08 |
190 |
68.84 |
71.96 |
197.03 |
196.69 |
|
IPD |
59.30 |
46.8 |
8.08 |
6.75 |
41.00 |
32.83 |
|
IPR |
431.15 |
393.93 |
72.17 |
69.99 |
286.75 |
265.86 |
|
PR |
39.93 |
16.18 |
10.94 |
4.62 |
23.05 |
9.37 |
|
MR |
78.93 |
68.88 |
69.72 |
64.54 |
59.84 |
52.99 |
|
MFS |
2.03 |
0.95 |
4.75 |
2.31 |
0.84 |
0.39 |
References
[1] Wei G, Wei F, Yuan C, et al. Integrated chemical and transcriptomic analysis reveals the distribution of protopanaxadiol- and protopanaxatriol-type saponins in Panax notoginseng. Molecules. 2018; 23(7):1773. Published 2018 Jul 19. doi:10.3390/molecules23071773.
[2] Qi X, Fang H, Yu X, Xu D, Li L, Liang C, Lu H, Li W, Chen Y, Chen Z. Transcriptome analysis of JA signal transduction, transcription factors, and monoterpene biosynthesis pathway in response to methyl jasmonate elicitation in Mentha canadensis L. Int J Mol Sci. 2018, 10;19(8):2364. doi: 10.3390/ijms19082364. PMID: 30103476; PMCID: PMC6121529.
[3]Yao H, Li C, Zhao H, Zhao J, Chen H, Bu T, Anhu W, Wu Q. Deep sequencing of the transcriptome reveals distinct flavonoid metabolism features of black tartary buckwheat (Fagopyrum tataricum Garetn.). Prog Biophys Mol Biol. 2017, 124:49-60. doi: 10.1016/j.pbiomolbio.2016.11.003. Epub 2016 Nov 9. PMID: 27836511.